# Testing the Scalability of the HS-AUTOFIT Tool in a High-Performance Computing Environment

**Giuseppe Di Modica** [1,*] **, Luca Evangelisti** [2] **, Luca Foschini** [1] **, Assimo Maris** [2] **and Sonia Melandri** [2]

[1] Dipartimento di Informatica, Scienze e Ingegneria, Università di Bologna, 40126 Bologna, Italy; luca.foschini@unibo.it
[2] Dipartimento di Chimica "G. Ciamician", Università di Bologna, 40126 Bologna, Italy; luca.evangelisti6@unibo.it (L.E.); assimo.maris@unibo.it (A.M.); sonia.melandri@unibo.it (S.M.)
[*] Correspondence: giuseppe.dimodica@unibo.it

**Abstract:** In the last years, the development of broadband chirped-pulse Fourier transform microwave spectrometers has revolutionized the field of rotational spectroscopy. Currently, it is possible to experimentally obtain a large quantity of spectra that would be difficult to analyze manually due to two main reasons. First, recent instruments allow obtaining a considerable amount of data in very short times, and second, it is possible to analyze complex mixtures of molecules that all contribute to the density of the spectra. AUTOFIT is a spectral assignment software application that was developed in 2013 to support and facilitate the analysis. Notwithstanding the benefits AUTOFIT brings in terms of automation of the analysis of the accumulated data, it still does not guarantee a good performance in terms of execution time because it leverages the computing power of a single computing machine. To cater to this requirement, we developed a parallel version of AUTOFIT, called HS-AUTOFIT, capable of running on high-performance computing (HPC) clusters to shorten the time to explore and analyze spectral big data. In this paper, we report some tests conducted on a real HPC cluster aimed at providing a quantitative assessment of HS-AUTOFIT's scaling capabilities in a multi-node computing context. The collected results demonstrate the benefits of the proposed approach in terms of a significant reduction in computing time.

**Keywords:** rotational spectroscopy; chirped-pulse Fourier transform; AUTOFIT; parallel computing; HPC; strong scalability; weak scalability

## 1. Introduction

In many fields of chemical science such as organic and inorganic synthesis [1] or analytical analysis [2], the unambiguous identification of the structure of molecules is of paramount importance.

Among the possible spectroscopic techniques developed in the past decades, Rotational spectroscopycan provide the most accurate measurements of molecular structures in the gas phase. The major technological advance in instrumentation in the field of rotational spectroscopy has been the development of the *chirped-pulse Fourier transform spectrometer* by Prof. Brooks H. Pate et al. [3] at the University of Virginia, which enables broadband spectrum acquisitions with unprecedented sensitivity. Improved sensitivity means high-density spectra and more precision in target spectra identification, but also much larger sets of data to analyze [4].

To analyze the complex spectra obtained from these measurements, the authors of [5] proposed an automated spectral assignment software application called AUTOFIT. The AUTOFIT algorithm takes advantage of the well-known benefits of the parallel computing paradigm and brings it into the rotational spectroscopy field. Specifically, the algorithm splits an entire spectrum into multiple pieces that can be analyzed in a parallel fashion by a number of "workers" (later, computing threads) in order to speed up the overall exploration. Unfortunately, AUTOFIT can only run on a single machine, therefore the

ability of the algorithm to scale up to more complex and broader spectra is bounded by the single machine's computing capacity, and specifically by the number of CPU cores in that machine.

To overcome this limitation, in a previous work [6] we proposed a new version of AUTOFIT (*HS-AUTOFIT*) capable of scaling out on multiple machines. This enhanced version of the application leverages the computing power of cluster nodes to offer theoretically unbounded horizontal scalability. We tested the viability of HS-AUTOFIT through preliminary experiments run on high-performance computing (HPC) and on cloud-computing environments.

In this paper, we report the results of quantitative tests run to assess the scaling capability of HS-AUTOFIT on an HPC cluster. For the purpose of the experiment, we made use of speedup performance metrics widely adopted in the literature [7,8], which helped us make reliable quantitative measurements.

Specifically, this paper extends our previous work [6] by

- providing insights on chirped-pulse Fourier transform spectroscopy;
- delivering a broader survey of the literature on spectral analysis techniques;
- proposing an extensive quantitative analysis of the scaling performance of HS-AUTOFIT in an HPC environment.

The rest of the paper is structured as follows. In Section 2, we review the recent literature in the field of rotational spectroscopy and introduce the metrics to assess the performance of parallel algorithms. In Section 3, we elaborate on the spectroscopy based on the chirped-pulse Fourier transform and the need for parallelizable software for fast and precise identification of molecules. In Section 4, we provide the reader with insights on the AUTOFIT software and introduce our proposal of a highly-scalable version of it (HS-AUTOFIT) that can leverage the computing power of HPC clusters. In Section 5, we measure the speedup that HS-AUTOFIT can gain in a real HPC environment. Finally, we conclude our work in Section 6.

## 2. Literature Review

In the past decades, several powerful spectroscopic techniques have been invented and developed to address the different challenges posed by the physical properties of molecules such as the state of matter, molecular structure, chirality [9], etc. Some of the most famous are nuclear magnetic resonance spectroscopy [10], X-ray [11] and, very recently, microcrystal electron diffraction (MicroED [12]). However, for small volatile molecules having permanent dipole moments, *rotational spectroscopy* can provide the most accurate measurements of molecular structures.

In the past, the analysis of rotational spectra was considered an art. Researchers manually analyzed the spectra by observing the flow of rotational transitions that appeared gradually in the experimental scan. By observing their positions and their intensities, they made hypotheses of assignment of the respective quantum numbers to classify the various transitions. This could result in an assignment of a conformational isomer with its respective rotational spectrum. However, this procedure is extremely time-consuming. Furthermore, the amount of data provided by the new chirped-pulse spectrometers makes this process inefficient as discriminating between the thousands of transitions currently observed in the rotational spectra is extremely difficult.

This has raised the need for computer tools capable of automation and speedup of the analysis of such data. Some of the most used approaches in the field are based on genetic algorithms [13], assignment via non-linear spectroscopy [14,15], implementation of artificial neural networks [16], robust automated assignment of rigid rotors [17], and algorithms for a broad search that fit all possible combinations of a set of transitions (such as AUTOFIT) [4]. Below, we report some technical details of the mentioned techniques.

Approaches based on *genetic algorithms* [13] analyze the rovibronic spectra for the direct determination of some molecular parameters, in particular the rotational constants. The algorithm input consists of an estimate of an acceptable range of certain spectroscopic

parameters. The theoretical spectra are evaluated and predicted using a rigid asymmetric rotor Hamiltonian. Afterwards, the genetic algorithm matches the generated theoretical spectra with the experimental spectrum.

Over the past years, various techniques based on *assignment via non-linear spectroscopy* [14,15] have been developed. The most relevant ones comply with the following procedure: (a) initially, the broadband spectrum of a molecule is acquired using classical chirped-pulse Fourier transform microwave spectroscopy and then it is analyzed by a combination of cavity Fourier transform microwave spectroscopy and double resonance; (b) to obtain the assignment of a conformer a strong field coherence breaking is used with the chirped-pulse Fourier transform microwave spectroscopy.

*Artificial neural networks* [16] are used to detect the typical patterns of the rotational spectra. Following the laws of quantum mechanics, thousands of rotational constants are generated to create rotational spectra. This resulting set is used as a training set to train an artificial neural network. The trained neural network is then employed to analyze an experimental spectrum. The algorithm was proved capable of identifying the type of spectrum (for example linear or asymmetric top) and deducing its spectroscopic parameters, e.g., rotational constants and hyperfine constants. Interestingly, the classification and prediction times appear to be independent of the spectral complexity.

The *robust automated assignment of rigid rotors (RAARR)* algorithm [17] can rapidly assign the experimental rotational spectrum for a "near-rigid" asymmetric top even in a mixture of conformers. In the first version of RAARR, $\mu_a$ and $\mu_b$ (the permanent electric dipole moment components along the *a* and *b* inertial axes) type lines are required to be non-zero. If such condition is met, the algorithm proceeds through the following steps: a search subroutine identifies rotational transitions in the spectrum; closely associated sets of lines are identified, in order to form a well-defined pattern of loops or series (called "scaffolds"); a set of *A*, *B*, and *C* rotational constants are obtained from each scaffold; starting from this attempt, a rotational spectrum is calculated, predicted, and assigned.

The original idea and implementation of the AUTOFIT algorithm was born in the laboratory of Prof. Brooks H. Pate. Originally, authors of AUTOFIT implemented the algorithm as a Mathcad script. The modern implementation is based on Python scripts and is due to Prof. Steven T. Shipman and Dr. Ian Finneran (https://github.com/pategroup/bband_scripts/tree/master/autofit, last access 2 February 2021). Recently, Shipman et al. [18] have also proposed a GRID-based AUTOFIT algorithm for removing as much user input as possible. Due to its inability to scale up, AUTOFIT is useless in cases where a large amount of input data are provided and a timely solution is requested.

In [6], we proposed HS-AUTOFIT, an enhanced and parallel version of the original AUTOFIT algorithm that exploits the computing power of multiple resources to speed up the completion time of highly demanding jobs. Specifically, we implemented two versions of HS-AUTOFIT that can execute on an HPC cluster and on Cloud-provided virtual machines, respectively. HS-AUTOFIT can be classified as an embarrassingly parallel application, as parallel tasks do not need to communicate with each other. Furthermore, it implements the *data parallelism* technique [19]; the input is partitioned into severaldata chunks that are then assigned to identical task instances for parallel elaboration.

In the literature, the main authoritative study on the scalability of parallel applications is due to Amdahl [7]. He pointed out that the speedup achievable for generic parallel software is limited by the fraction of the serial part of the software that is not amenable to parallelization. Amdahl mostly focused on the performance index widely known as *strong scaling*, which defines how the computation time varies with the number of processors for a fixed total problem size. A few years later, Gustafson proposed a new perspective of the scalability index that combines the increase in cores with the increase in the input load [8]. Such index is known as *weak scaling*, and defines how the computation time varies with the number of processors for a fixed problem size per processor.

Strong scaling and weak scaling indices are widely recognized in the literature as the most reliable metrics to provide quantitative measures of the speedup that applications can

gain in parallel computing contexts. We will stick to those definitions to assess the scaling performance of HS-AUTOFIT.

## 3. Background and Motivation

Molecular structure has been one of the most important pieces of information obtainable from the analysis of the *rotational spectrum*. This has allowed the study of different molecules and molecular weakly bound complexes, thus characterizing different types of chemical bonds [20]. The major instrumentation advance in the field of rotational spectroscopy in the last years has been the *chirped-pulse Fourier transform spectrometer* [3]. It enables broadband spectrum acquisitions with high sensitivity. Until that invention, the previous instruments were dominated by the Balle–Flygare design, which required the incorporation of a cavity resonator [21]. In this case the cavity works like a "resonant circuit" with a high quality factor $Q$, which means a lower rate of energy loss but a narrow measurement bandwidth (about 1 MHz). For many years these types of spectrometers have remained similar to the original version [22] and even the modern versions [23] have an average scanning speed (due to the different experimental conditions of use) of about 1 GHz per day. However, with this technique the rotational spectrum of typical analyzed molecules spans over several GHz. The development of the new chirped-pulse spectrometer was made possible thanks to the implementation of high-speed digital electronics. In fact, the instrument uses a high-speed arbitrary waveform generator to generate the excitation pulse (the so called chirped pulse) and a high-speed oscilloscope to digitize the molecular free induction decay. The oscilloscopes used, for example, remain a fundamental tool for the development of other important technologies. Therefore, their technological development is very important and there is intense research activity. Starting around 1993, a law similar to Moore's law has been observed for arbitrary waveform generators and oscilloscopes. The law dictates that device bandwidth doubles every three years [24]. This technological advancement makes it possible to acquire an ever greater quantity of data in less and less time. In this sense, the development of automatic analysis tools that can be scaled on an HPC system is essential to allow the data to be analyzed in a reasonable time. The new chirped-pulse Fourier transform spectrometers are able to record a large bandwidth (around 12 GHz), so the full operating range of the spectrometer is obtained on each measurement cycle [25]. The result is an instrument that allows one to obtain very high sensitivity and dense spectra that can show an average line density of over 1 MHz$^{-1}$.

For decades, rotational spectroscopy has been widely used for the study of the chemical composition of interstellar media [26]. The comparison between the rotational transitions measured in the laboratory and those observed by radio telescopes allows a reliable identification of the molecules present in the interstellar medium. Moreover, very recently the technique has proved to be of fundamental use in the field of analytical chemistry as a support method for various scientific fields, for example in pharmaceutical research [27] such as in the analysis of isotopic impurities [28]. The ultimate goal in the field is to be able to create a technique that has the potential to directly analyze complex chemical mixtures and also perform chiral analyses without the need for chemical separation by chromatography [29,30]. The reader may find more insights on the on the chirped-pulse Fourier transform spectrometer in Appendix A.

In order to pursue these objectives, the analysis of the spectra, and in general of the data obtained through this technique, must be made fast, simple, and automated so that even non-specialized personnel can use it and extract the information in a reasonable time.

In this work, we discuss the fitting method based on the AUTOFIT algorithm. The success of the algorithm is demonstrated by several microwave works based on the use of this algorithm and some other methods use a similar philosophy [31,32]. This is due to several reasons: the Hamiltonian for these systems is known; very accurate estimates for the rotational constants (and spectroscopic parameters in general) are possible through quantum chemical calculations; and the frequencies of the rotational transitions are determined by the Hamiltonian to experimental accuracy. Moreover, rotational spectroscopy is

an intrinsic high-resolution technique and the root-mean-squared frequency rest between predicted and experimental transitions is of the order of 10 kHz.

Detecting a molecule within an experimental spectrum requires exploring many combinations of triplets. The higher the requested goal accuracy, the higher the number of algorithm iterations necessary to achieve the goal. In particular, the number of iterations is proportional to the size of the spectrum containing the transitions; therefore, in current experiments, typically a few thousands to millions of combinations need to be examined. On average, AUTOFIT can analyze up to 35–50 triplets per second per CPU core, therefore on a typical 4-core processor it will reach a triplet processing rate of around 250 per second. For instance, if we consider a frequency window of 200–300 MHz, AUTOFIT will take from 2 to 3 h to complete the job. Modern spectrometers can deliver a huge quantity of data that translate into very dense search windows for which AUTOFIT cannot guarantee an acceptable execution time, thus making the tool useless.

Motivated by this compelling need, we developed a new strategy aimed at accommodating any level of accuracy requested while ensuring a much shorter execution time. The proposed approach leverages the parallel computing paradigm to exploit the speedup power provided by multiple computing resources to explore the solution space in a fast way.

## 4. HS-AUTOFIT: A Highly Scalable AUTOFIT Application

In this section, we firstly touch on the AUTOFIT basic approach [4]; then, we discuss HS-AUTOFIT, an enhanced version of the AUTOFIT algorithm that leverages the computing power of a cluster of nodes in order to sustain the scaling performance required by highly demanding jobs.

In Figure 1, we show the basic steps of the AUTOFIT algorithm. To model the algorithm, we adopted the UML activity diagram notation that allows us to represent parallel computing flows that branch off at some point in the main flow. In the diagram, ellipses represent computing steps, while boxes contain either input or output data that go to, or are produced by, computing steps. Solid arrows define the sequencing order of the steps, whereas dashed arrows indicate which data are associated to which computing steps (as either input or output). The two flat boxes model synchronization points where multiple parallel computing flows branch off and merge, respectively. In the figure, for space reasons we have depicted only two parallel flows branching off the main workflow (namely, *Worker*1 and *Worker*2 flows).

The core of the algorithm and the procedure are similar to the original one [4]. Input to the AUTOFIT algorithm are the rotational constants *A*, *B*, and *C*, a set of dipole moment components ($D_m$)—predetermined via quantum chemical calculations—and some experimental parameters such as the limits on the intensity of the transitions ($I_t$), the possible nuclear quadrupole constants ($Q_i$) and the rotational temperature ($K_r$) of the analyzed molecular sample. From these data, the SPFIT and SPCAT tools bundled with the CALPGM suite [33] will be used to predict and fit the spectra. From the transition predicted in the experimental frequency range, it will be necessary to select the triplet to look for ($T_s$) and a set of transitions that are used for scoring the obtained fits from the triplets. For each frequency in the chosen triplet, a frequency interval ($f_i$) must also be provided. From these intervals, the algorithm will identify in the input microwave data set (the experimental spectrum ($S$)) all the possible lines that have to be used to search for useful triplets.

According to the AUTOFIT design, the overall dataset is split into multiple, smaller data chunks that independent workers will explore to find the fitting triplets. Then, AUTOFIT will fit all possible combinations of triplets within the range window of the observed spectrum. Once assigned certain experimental frequencies in the subset ($S_i$) to the triplet, SPFIT calculates the rotational constants and predicts the theoretical frequency of the control transitions (*Trans*). The theoretical prediction of the control transitions is compared with the closest transitions observed in the experimental spectrum. Using a least-squares method for the error between the experimental frequencies measured and those theoretically calculated from the rotational constants obtained, SPFIT provides a total

value on the error associated with the fit. This is called RMS [34] and is used to evaluate how good the fits are and to build the scoring function.

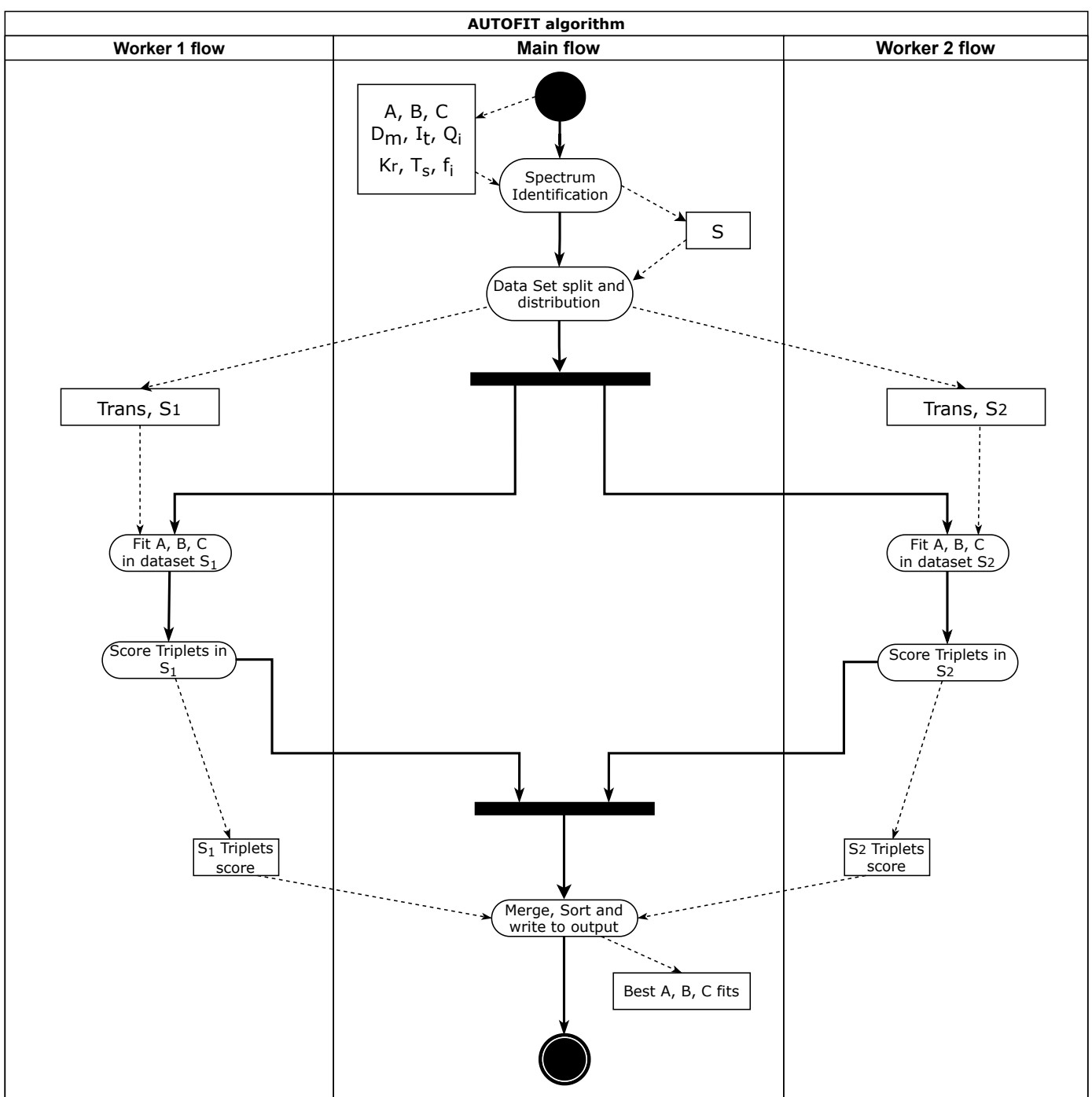

**Figure 1.** The AUTOFIT algorithm.

From a computing perspective, the triplet fitting step is an embarrassingly parallelizable job. The entire spectrum data to explore can be split into multiple data chunks ($S_i$), each of which is then assigned to an independent *search task*, which does not need to communicate with its peers to carry out its own work. This kind of parallelism is known in the literature as *data parallelism* [19]. Eventually, AUTOFIT collects results from all task computation and merges them in the final output. By virtue of the described approach, AUTOFIT can exploit the CPU cores of a computing machine (even an off-the-shelf PC) in order to shorten the overall computation time. To give an explanatory example, on an *N*-core CPU

we could split the spectrum data into *N* equally-sized data chunks, and feed them to as many search tasks that will execute in the form of *computing threads* (each served by one CPU core). Unfortunately, increasing the number of data chunks (and consequentially, of threads) beyond the number of CPUs, available cores will not produce a performance gain, as the exceeding number of threads would still have to wait for free cores to execute. For this reason, AUTOFIT cannot accommodate highly demanding jobs. If the theoretical results are not as accurate as requested, the frequency range in which to look up fitting triplets needs to be widened. A wider spectrum will produce a much larger dataset (*S*, in Figure 1) which, given the limited number of available cores, can cause an unacceptable increase in the AUTOFIT execution time, thus making the tool useless.

The main AUTOFIT flaw is the inability to scale over the number of a computing node's CPU cores. We propose to overcome such a limitation by leveraging the scale power typically offered by dense computing contexts exploiting HPC technology. To this end, in [6] we proposed HS-AUTOFIT, a modified version of AUTOFIT capable of scaling out to multiple CPU cores of an HPC cluster of nodes. Specifically, to support the distribution of tasks among the cores of an HPC cluster, HS-AUTOFIT offers the following services: transfer of generic data types and small-sized data (e.g., configuration parameters); transfer/sharing of text files (containing the triplets to fit); and support for a synchronized communication mode. We adopted the *message passing interface (MPI)* protocol to implement low-level communication among HPC nodes. The MPI service is provided by common software libraries (https://mpi4py.readthedocs.io/en/stable/index.html, accessed on 30 June 2021) developed in Python 2.7, which is the programming language that was originally used to code the AUTOFIT application, to which we applied the aforementioned modifications.

## 5. Experiment

In this section, we discuss the experimental run to assess the scaling capability of the HS-AUTOFIT algorithm on the GALILEO HPC cluster hosted at CINECA [35]. GALILEO is a hybrid HPC cluster. Out of its 1084 nodes, 1022 are equipped with an Intel Xeon E5-2697 v4, 60 with an nVidia K80 GPU and 2 with an nVidia V100 GPU. Table 1 shows the main technical features of the GALILEO cluster.

The *Slurm* workload manager is in place in CINECA. Slurm is an open-source and highly scalable cluster management and job scheduling system. As per the CINECA policy, restrictions are applied to job computing requests issued by users through Slurm. In particular, the following limitation held for our experiments: each job can be assigned a maximum of 256 cores/4 nodes and a maximum memory capacity of 118 MB per node. While communication of small parameters among the master workflow and the workers was implemented through MPI, for the sharing of consistent input data (such as, e.g., the frequency spectrum to explore) we leveraged the general parallel file system (GPFS) provided by the CINECA cluster. In order to mitigate the uncertainty over obtained results (due to both the scheduler choices and the variability of the overall workload to be managed by the cluster), we repeated each test 20 times and eventually considered average results.

Finally, every node involved in the experiments was equipped with a Python engine and all necessary software libraries, as well as the fundamental tools needed by HS-AUTOFIT to execute. The objective of the experiments was to assess the performance of HS-AUTOFIT in terms of both strong and weak scaling capabilities. The experimental design and test results are discussed in the following sections.

**Table 1.** Main features of the GALILEO HPC cluster.

| Model | IBM NeXtScale cluster |
|---|---|
| Architecture | Linux Infiniband cluster |
| Network | Intel OmniPath (100 Gb/s) high-performance network |
| Nodes | 1022 Intel Broadwell<br>60 Intel Broadwell with GPUs<br>2 Intel Broadwell with GPUs |
| Processors | 2 × 18-cores Intel Xeon E5-2697 v4 at 2.30 GHz (26,572 cores in total)<br>2 × 18-cores Intel Xeon E5-2697 v4 at 2.30 GHz + 2 nVidia K80 GPUs (2160 cores in total)<br>2 × 18-cores Intel Xeon E5-2697 v4 at 2.30 GHz + 2 nVidia V100 GPUs (72 cores in total) |
| RAM | 128 GB/node |

### 5.1. Strong Scaling Test

In order to measure the *strong scaling* performance of an application, we consider the case when the problem size is fixed but the number of processing elements is gradually increased. The higher the number of processors, the lower the assigned workload per processor. An application is said to scale linearly if the speedup (in terms of work units completed per unit time) is equal to the number of processing elements used (N). In general, linearity is a hard goal to attain as with larger process counts the communication and I/O overhead grow and jeopardize the overall performance [7].

Let $t_1$ be the amount of time needed to complete a work unit as a serial task, and $t_N$ be the amount of time to complete the same unit of work with N processing elements (parallel tasks). *Speedup* is defined as

$$Speedup_N = \frac{t_1}{t_N} \tag{1}$$

In the case of HS-AUTOFIT, we fixed the problem size by setting the frequency of the search window to 200 MHz across all tests. Such frequency produced an input of around 253,456 triplets, among which the fitting triplets need to be discovered. We then ran nine consecutive tests to assess the speedup trend of HS-AUTOFIT on 1, 2, 4, 8, 16, 32, 64, 128, and 256 processing core elements. The objective of the test is to derive the execution time of the parallel portion of the entire algorithm flow. In Figure 1, parallel flows start when the master flow distributes data to the workers, and ends when all workers have returned the output of their elaboration to the master flow.

In Figure 2, we depict the trend of HS-AUTOFIT strong scaling speedup along with the curve of the ideal speedup. The reader may notice that the trend is linear up to 4 processing cores. From 8 cores onward, although still increasing, the speedup follows a parabolic trend.

Strictly correlated to the speedup index is the *parallelism efficiency*, which in the case that *N* core processors are used, is defined as

$$efficiency_N = \frac{t_1}{t_N \times N} \tag{2}$$

In the ideal case, efficiency is a straight line with a constant value of 1. In the graph of Figure 3, both ideal and HS-AUTOFIT efficiency are depicted.

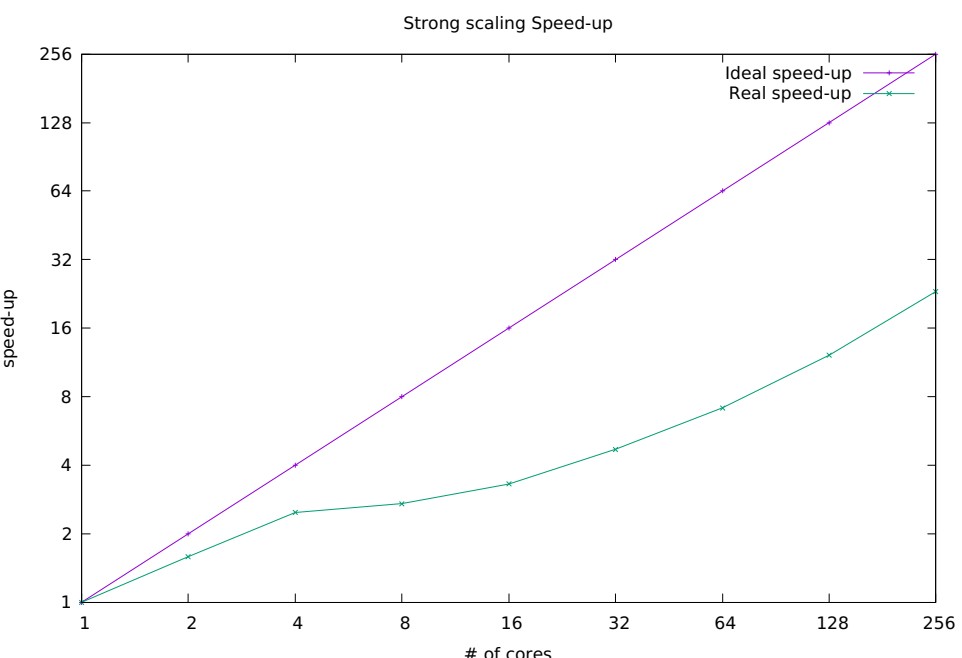

**Figure 2.** Strong scaling speedup.

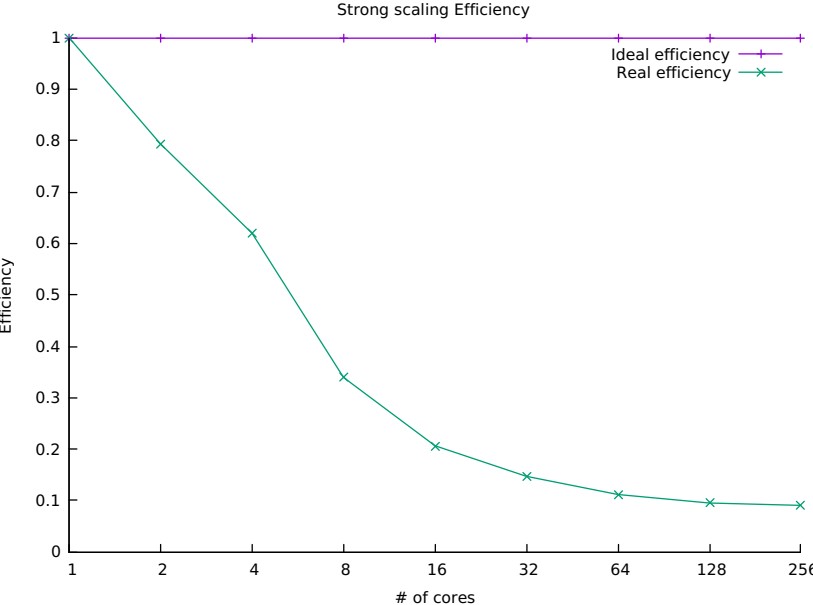

**Figure 3.** Strong scaling efficiency.

From the obtained results, we can conclude that the application can scale quite well, although its parallel efficiency is far from ideal. There are some factors that impact negatively on the strong scaling metric. To cite one, HS-AUTOFIT is an I/O bound application. It frequently accesses the storage, which introduces a lot of delay in computation. In addition, CINECA is a very busy system built upon one shared file system (GPFS), which of course cannot always guarantee a constant performance in terms of data fetch speed. In our future work, we will improve the algorithm to reduce access to persistent storage and make more use of cores assigned memory.

### 5.2. Weak Scaling Test

The *weak scaling index* provides a measure of the speedup performance that the application can gain from leveraging the power of an increasing count of processing cores,

provided that the overall workload increases so that every added core gets the same, fixed-size workload. For this experiment, we considered a reference working unit of 8000 triplets. We ran nine tests to assess the speedup trend on 1, 2, 4, 8, 16, 32, 64, 128, and 256 processing cores, taking care that each core was assigned a workload size that equaled that of the reference working unit (8000 triplets). In practice, at each step we doubled both the core count and the overall workload. The overall assignment of cores/triplets for the nine tests is depicted in Table 2.

**Table 2.** Core/triplet assignment in the weak scaling experiment.

| #cores | 1 | 2 | 4 | 8 | 16 | 32 | 64 | 128 | 256 |
|---|---|---|---|---|---|---|---|---|---|
| triplets | 8000 | 16,000 | 32,000 | 64,000 | 128,000 | 256,000 | 512,000 | 1,024,000 | 2,048,000 |

For this experiment, we borrow the parallelism efficiency definition of Equation (2). Results are depicted in Figure 4. Unfortunately, efficiency degrades very close to the value 0.2 (#cores 8), then it keeps quite steady to the increase of the core counts. As mentioned earlier, access to the storage is still a weak point of HS-AUTOFIT, which we intend to improve in the next release. We are confident this could improve the current figures. With that said, we claim that results obtained in this work can guide researchers to look for the optimal core configuration that best suits their need.

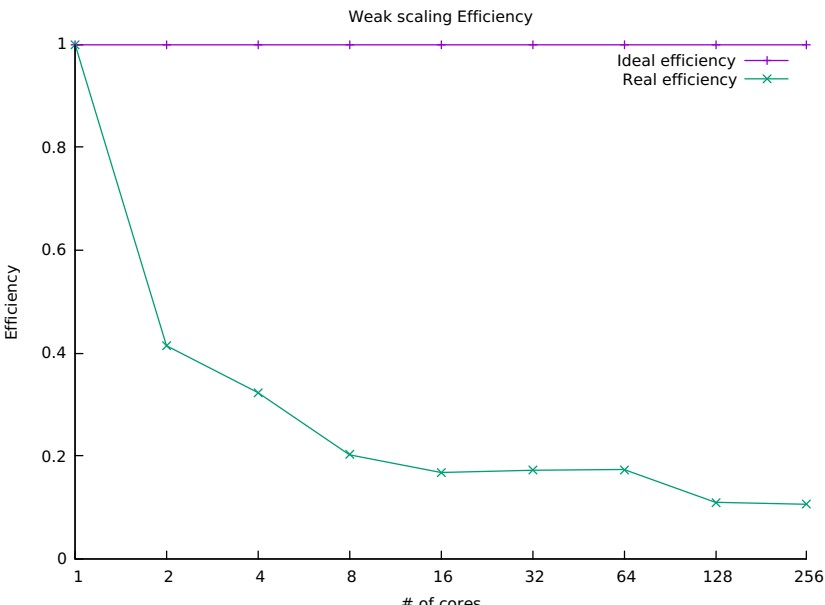

**Figure 4.** Weak scaling efficiency.

*5.3. Statistical Significance of the Experiment*

Each of the timing values used in the speedup Formulas (1) and (2) are obtained from averaging the results of 20 tests. In this section, we provide some information concerning the statistical characterization of the data gathered in the strong scaling test.

In the tests, we considered the following workloads: 7957 triplets, 39,556 triplets, and 231,880 triplets. For each workload, we ran HS-AUTOFIT instances with increasing numbers of cores in the range [1–16]. In Figure 5a,b, we report the results of the tests in the form of box-and-whisker charts for the workloads of 7957 triplets and 231,880 triplets, respectively. The chart depicting the results for the 39,556 triplet configuration was omitted for space reasons. The reader will notice that the higher the workload, the lower the uncertainty of obtained results. Furthermore, for both workload configurations, the result uncertainty decreases as the core counts increase. We can conclude that, in the case

of consistent workloads and a relatively high number of cores, the results have robust statistical significance.

Finally, the benefits of parallelization are clear for the 231,880 triplets workload: the trend shows that the algorithm execution times get shorter as more cores are employed. This is also true in the case of 7957 triplets, although when the core counts increased from 2 to 4 no clear gain was observed.

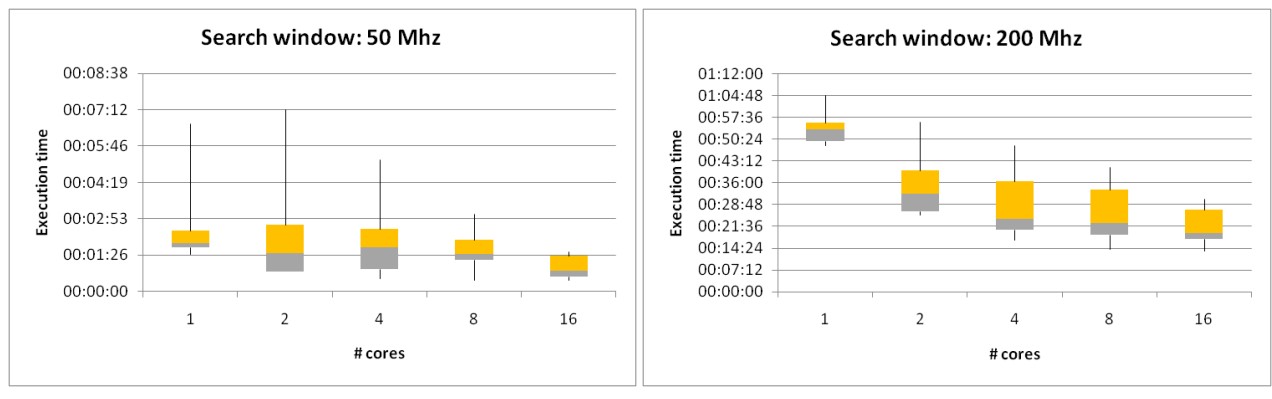

(**a**) Workload: 7957 triplets        (**b**) Workload: 231,880 triplets

**Figure 5.** Statistical significance of the experiments.

## 6. Conclusions and Future Work

Being able to fit microwave spectra is an important as well as a complicated task. The difficulty in manually fitting the spectra has resulted over the years in the development of software that has helped in this intent. AUTOFIT has proved to be effective for both routine analysis of pure samples and the analysis of mixtures, which are becoming increasingly important. Yet, AUTOFIT's performance (computing time) strongly depends on the number and on the density of rotational transitions measured in the spectrum. The improvement in the electronics and the desire to analyze increasingly complex mixtures has increased over the years the amount of data that can be recorded and that must be analyzed. To meet this requirement, we have developed a new version of the algorithm (HS-AUTOFIT) that leverages the computing power of a multi-node computing infrastructure to speed up the overall computing time. In a previous work, we showed that HS-AUTOFIT can provide a quicker response time when deployed both in an HPC and in a Cloud runtime context. In this paper, we have assessed the scaling capabilities of HS-AUTOFIT through extensive tests conducted in a real HPC cluster.

Regarding future development of the present work, we will aim to improve HS-AUTOFIT's overall efficiency by reducing the storage access rate. We believe this will also bring benefits in terms of both strong and weak scaling. In addition, the savings in analysis time can be exploited for an implementation of the fitting procedure. Currently a triplet is used to derive the three rotational constants $A$, $B$, and $C$. In future developments, the efficiency of the algorithm will allow us to use a quadruplet with better prospects to obtain reliable results. Furthermore, AUTOFIT has a restriction due to the ability to fit only "rigid rotor molecules", which precludes excellent results when the flexibility of the molecules is higher or shows hyperfine patterns due to nuclear coupling effects (such as those produced by $^{14}N$ in organic molecules). Although the observed splitting is generally not very large, the results obtained are not as good as those for molecules without quadrupole atoms. The situation becomes even more complicated when the molecules contain atoms such as chlorine, bromine, iodine, etc., for which the quadrupole splits can reach different MHz. In this implementation, an option has been included in the input file to specify the quadrupole parameters that can be obtained from ab initio calculations. This ability allows us to fit molecules containing atoms with quadrupoles obtaining RMS errors of the fit comparable to common "rigid rotor molecules".

**Author Contributions:** Conceptualization, G.D.M., L.E., L.F., A.M. and S.M.; methodology, G.D.M., L.E., L.F., A.M. and S.M.; software, L.E.; validation, G.D.M., L.E.; formal analysis, L.F., A.M. and S.M.; investigation, G.D.M., L.E.; writing—original draft preparation, G.D.M., L.E.; writing—review and editing, L.F., A.M. and S.M. All authors have read and agreed to the published version of the manuscript.

**Funding:** This research received no external funding

**Data Availability Statement:** Not applicable.

**Conflicts of Interest:** The authors declare no conflict of interest

## Abbreviations

The following abbreviations are used in this manuscript:

| | |
|---|---|
| RAARR | Robust automated assignment of rigid rotors |
| HPC | High-performance computing |
| HS-AUTOFIT | Highly scalable AUTOFIT |

## Appendix A. The Chirped-Pulse Fourier Transform Spectrometer

There are several excellent textbooks that cover the theory of rotational spectroscopy (also called microwave spectroscopy due to the frequency range of the technique) [36–38]. In brief, the technique requires the molecules to be in the gas phase during the analysis. The energy levels of the molecular systems come from the kinetic energy of rotation and since the angular momentum is quantized, the allowed energies for the molecular rotation are quantized. These energies can be obtained from the eigenvalues of the Hamiltonian operator and are related to the moments of inertia ($I$) of the molecules (and so to the molecular structure). The majority of modern spectrometers work in supersonic expansion so the rotational temperature of the molecules is very low (typically 1–10 K) and this allows the molecules to be approximated as rigid rotors. For a specific molecule, there are no more than three different moments of inertia ($I_a$, $I_b$, $I_c$) and the molecules can be classified based on the relative values of these parameters, which in turn are related to the symmetry of their structure:

- Spherical top molecules

$$I_a = I_b = I_c \tag{A1}$$

- Linear molecules

$$I_a = 0, I_b = I_c \tag{A2}$$

- Symmetric top molecules

$$I_a = I_b < I_c \quad \text{or} \quad I_a < I_b = I_c \tag{A3}$$

- Asymmetric top molecules

$$I_a < I_b < I_c \tag{A4}$$

The last class is the most important since it is the most common case. In this case, for a rigid rotor, the rotational transition frequencies are determined only by the rotational constants $A$, $B$, and $C$, which are related to the moment of inertia by the following equations:

$$A = \frac{\hbar}{2I_a} \qquad B = \frac{\hbar}{2I_b} \qquad C = \frac{\hbar}{2I_c} \tag{A5}$$

For a given set of $A$, $B$, and $C$ rotational constants, it is now very easy to predict the microwave spectrum using the Watson asymmetric top Hamiltonian [39]. Several software packages such as Pickett's program [33] are available for this purpose [40–42]. A more time-consuming procedure is required for solving the opposite problem, which is the assignment of a microwave spectrum of a molecule and the determination of its $A$, $B$, and $C$ rotational constants. For doing this it is necessary to assign the correct labeling (the quantum

numbers) to the lines in the recorded spectra. However, using a modern chirped-pulse microwave spectrometer, each spectrum comprises thousands of lines and several molecules and molecular complexes are simultaneously observed. For example, Figure A1 shows the rotational spectrum of difluoromethane, which has recently been published [43,44].

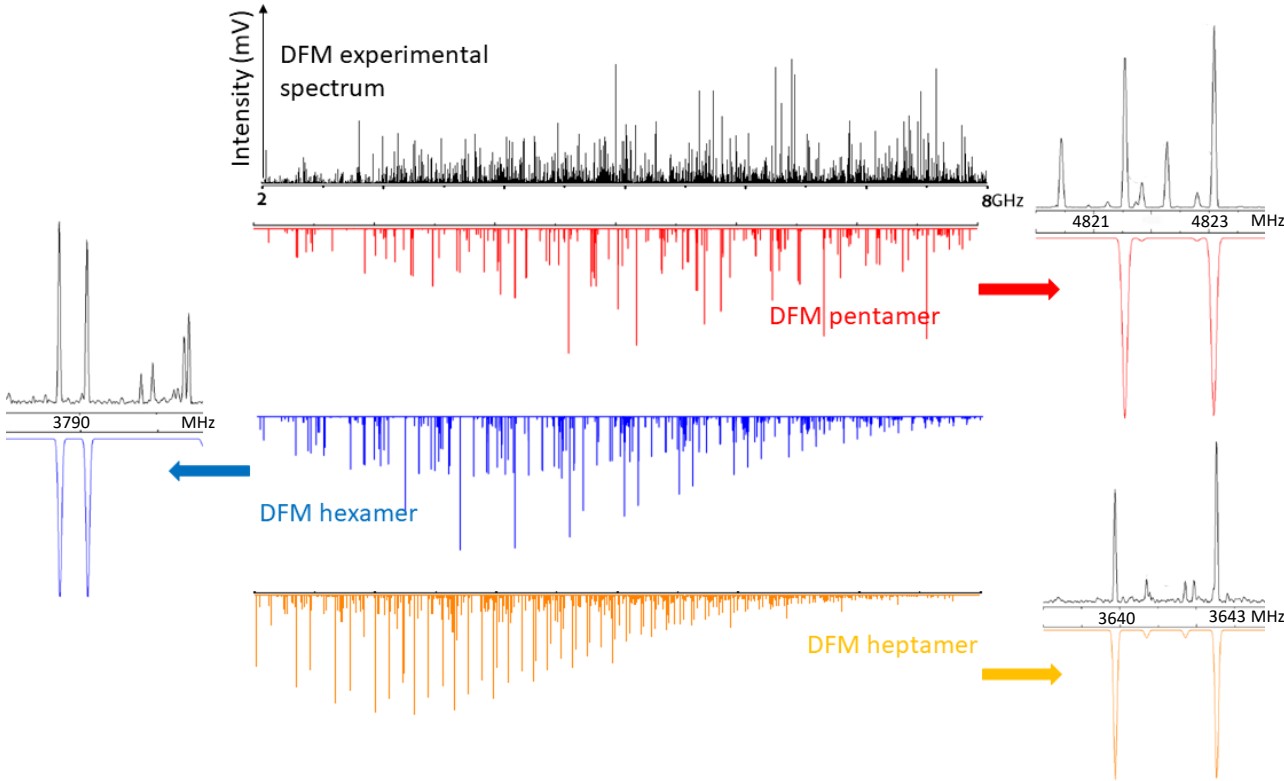

**Figure A1.** The 2–8 GHz spectrum of difluoromethane (top black), with predictions of three assigned clusters. Specifically, red transitions correspond to difluoromethane pentamer, blue to difluoromethane hexamer, and orange to difluoromethane heptamer. The rotational temperature chosen for the predictions was 2 K, which best reproduces the spectral intensities for the clusters, whilst the relative intensities between the three theoretical spectra are not to scale.

The black spectrum at the top is the observed experimental spectrum. In this same spectrum different isomers have been observed, each characterized by its own rotational spectrum, which is like a fingerprint characteristic of that particular species. Among the various isomers observed, three of them are shown in red, blue, and orange. They refer to the pentamer, a hexamer, and the heptamer of the difluoromethane clusters, respectively. This makes the spectra really dense and the assignment of a rotational transition belonging to a specific molecule is not a trivial task. These spectra are the result of fitting the various spectroscopic parameters. It can be seen as shown on the right and left of the various spectra that there is a perfect correspondence between the experimental rotational transitions and the theoretical prediction obtained from the fitting of the rotational constants. The accuracy between measurement and theory is very high (a few kHz).

Currently, to perform a spectral assignment we rely on two distinct steps. First of all, rotational constants are predicted using modern quantum chemical calculations [45]. Several commercial [46–48] or free [49–51] software packages are available. They allow us to perform state-of-the-art quantum chemical simulations and predict the rotational constants with typical errors of less than 1% between the calculated values and the final experimental values. Guided by the theoretical results, the experimental spectrum must then be inspected in search of the individual rotational transitions. Typically, a simulated spectrum is set using the ab initio calculations and is directly compared to the experiment. If the spectral pattern is relatively obvious, then the matching between the theoretical prediction and the experiment is straightforward and the assignment can be performed

manually [52]. Interactive programs have been created to facilitate the procedure [31,53,54]. It is quite easy for linear and symmetric top molecules. The assignment becomes more difficult for asymmetric tops because the patterns are more complicated. In this case, computer assistance could become fundamental for successful analysis. It is in this sector where the use of automated algorithms becomes of paramount importance to speed up the identification of rotational spectra.

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
