# Peer review of "Testing the Scalability of the HS-AUTOFIT Tool in a High-Performance Computing Environment"

_electronics, doi:10.3390/electronics10182251_

Round 1

Reviewer 1 Report

The authors presented some results of tests performed to study the scaling capability of an improved AUTOFIT application.

The work is well presented. However, some questions:

1) The authores say HS-AUTOFIT can provide quicker response time when depliyed both in an HPC and in a Cloud runtime context. However the experiment only talks about the results on an HPC cluster. Please, comment about this.

2)  In line 286 authors say there are some factors that impact negatively on the strong sacling metric. However, they only talk about one factor. Can the authors comment on some other factors?

Author Response

The authors presented some results of tests performed to study the scaling capability of an improved AUTOFIT application. 

The work is well presented. However, some questions: 

C1.1: 1) The authores say HS-AUTOFIT can provide quicker response time when depliyed both in an HPC and in a Cloud runtime context. However the experiment only talks about the results on an HPC cluster. Please, comment about this. 

R1.1: In a former work of ours (Corradi et al., 2020) the reviewer may find results from preliminary experiments conducted to test HS-AUTOFIT in a Cloud environment. That work was aimed at proving that HS-AUTOFIT can fit both computing facilities specialized for scientific applications, such as the HPC, and more general-purpose computing environments like the Cloud. In this work, we investigate the HS-AUTOFIT capability of scaling up to highly demanding jobs. To that purpose, we opted for the HPC as testbed for our experiments as it guarantees a better stability for what concerns resource performance (like e.g., inter-node bandwidth). 

C1.2: 2) In line 286 authors say there are some factors that impact negatively on the strong sacling metric. However, they only talk about one factor. Can the authors comment on some other factors? 

R1.2: In the paper, we actually cited two factors: 1) application’s I/O boundness; 2) instability of CINECA GPFS performance. We re-edited the paragraph to make this much clearer.

Reviewer 2 Report

The manuscript lacks scientific merit. In particular, the problem is unclear, the proposed method is straightforward, and the novelty of the proposed method is limited.

The manuscript is like a conference-level paper (although the authors claim that it is an extension of their previous conference paper).

Author Response

C2.1: The manuscript lacks scientific merit. In particular, the problem is unclear, the proposed method is straightforward, and the novelty of the proposed method is limited. 
The manuscript is like a conference-level paper (although the authors claim that it is an extension of their previous conference paper). 

R2.1: We thank the reviewer for the comment, which we address with the answer below.  
Concerning the problem, in Section 3 (lines 139 -- 162) we enhanced the discussion on the current need, as well as the scientific gap, that motivated our proposal. Specifically, our big-data tool was designed to meet the need of elaborating in a timely fashion the big amounts of data produced by new generation chirped pulse spectrometers. As reported in the paper, this is a real and compelling need in many fields such as pharmaceutics and astrochemistry, which so far has not yet been addressed by the research community. While in a former conference paper (Corradi et al., 2020) we reported preliminary results on the viability of the proposed algorithm in distributed computing environment, in this manuscript we study and report on the scalability feature of the algorithm itself. 
We hope this helps clarify the reviewer’s concern. Should more specific points be raised by the reviewer, we will be happy to provide justification.

Reviewer 3 Report

This paper presents the improved version of the AUTOFIT tool, called HS-AUTOFIT, which is a spectral assignment software application capable of running on High Performance Computing clusters in order to shorten the time to  explore and analyze spectral big data. Further, it reports the results of the tests run to assess the scaling capability of HS-AUTOFIT. Also, it describes its limitation, which is running on a single machine and depending on its CPU cores.

The article presents the scalability testing phase of the Highly Scalable AUTOFIT (HS-AUTOFIT) tool in a High Performance Computing (HPC) environment. AUTOFIT is a spectral assignment software application created in 2013 to contribute and facilitate the analysis of microwave spectra. Applying the manual analysis to a large quantity of spectra obtained through experimental methods would be challenging. HS-AUTOFIT is another version of AUTOFIT that explores spectral big data faster than AUTOFIT. Some tests have been detailed in the paper on a real HPC cluster to provide a quantitative assessment of HS-AUTOFIT’s scaling capabilities in a multi-node computing context.

This paper has a good structure. The diagrams, graphs, and tables have relevant titles and good sizes. The presented formulas use a suitable font and are well-positioned regarding the paper's template. Figure 1 looks very good and helps in understanding the main steps of the AUTOFIT algorithm. There is a short subsection for abbreviations; I think it should've appeared at the beginning of the paper.

The authors should be paying more attention to the spelling check and grammar rules: “recent instruments allow to obtain considerable amount of data” - recent instruments allow obtaining a considerable amount of data “To cater for this requirement”- To cater to this requirement “ Collected result demonstrate the benefits”- The collected result demonstrates the benefits “In many fields of the chemical science”- In many fields of chemical science “high density”- high-density “the algorithm splits the entire spectra in multiple pieces”- the algorithm splits the entire spectra into multiple pieces Some references, especially those from 19xx, need to be updated with more recent work (i.e. 2021). 

Author Response

This paper presents the improved version of the AUTOFIT tool, called HS-AUTOFIT, which is a spectral assignment software application capable of running on High Performance Computing clusters in order to shorten the time to explore and analyze spectral big data. Further, it reports the results of the tests run to assess the scaling capability of HS-AUTOFIT. Also, it describes its limitation, which is running on a single machine and depending on its CPU cores. 

The article presents the scalability testing phase of the Highly Scalable AUTOFIT (HS-AUTOFIT) tool in a High Performance Computing (HPC) environment. AUTOFIT is a spectral assignment software application created in 2013 to contribute and facilitate the analysis of microwave spectra. Applying the manual analysis to a large quantity of spectra obtained through experimental methods would be challenging. HS-AUTOFIT is another version of AUTOFIT that explores spectral big data faster than AUTOFIT. Some tests have been detailed in the paper on a real HPC cluster to provide a quantitative assessment of HS-AUTOFIT’s scaling capabilities in a multi-node computing context. 

C3.1: This paper has a good structure. The diagrams, graphs, and tables have relevant titles and good sizes. The presented formulas use a suitable font and are well-positioned regarding the paper's template. Figure 1 looks very good and helps in understanding the main steps of the AUTOFIT algorithm. There is a short subsection for abbreviations; I think it should've appeared at the beginning of the paper. 

R3.1: We thank the reviewer for the useful hint. As per the journal template, the abbreviation section must appear where it has been currently placed. 

C3.2: The authors should be paying more attention to the spelling check and grammar rules: “recent instruments allow to obtain considerable amount of data” - recent instruments allow obtaining a considerable amount of data “To cater for this requirement”- To cater to this requirement “ Collected result demonstrate the benefits”- The collected result demonstrates the benefits “In many fields of the chemical science”- In many fields of chemical science “high density”- high-density “the algorithm splits the entire spectra in multiple pieces”- the algorithm splits the entire spectra into multiple pieces Some references, especially those from 19xx, need to be updated with more recent work (i.e. 2021). 

R3.2: We thank the reviewer for the grammar suggestions, which we considered in the final proof-reading. Regarding the references dated in 19xx (specifically refs .: 11, 22, 38, 40, 41, 42, 50) we cannot update them with more modern references (year 20xx) as they represent the basic theory on which the technology is based and still represent the state of the art on the subject.

Reviewer 4 Report

The authors have to accurately address the below comments.

  • Paper title: It prefers to replace the abbreviations (such as HPS ...) in the search title with words to be clearer to the readers and researchers.
  • The "Literature review" section contains a lot of useful information, however, the authors point out some references that used "AUTOFIT" but they did not clearly state drawbacks of these papers.
  • The authors presented their findings in an appropriate structured way, and they also provide some flaws in their proposed nicely. However, the authors did not provide comparisons between their findings and the results of existing research to support rationality and fairness in the superiority of their proposed (despite the different environments and parameters).
  • Figures and Tables: All figures and tables are drawn in high resolution. However, Figures 4, 5 and Tables 1, 2 shown in-text before use.
  • Paraphrase text: There are many sentences taken from previous papers. Authors must paraphrase all sentences and phrases taken from the previous papers. The authors must avoid taking whole sentences from the original papers even if it is for the same authors. They are obligated to paraphrase during the revision of the entire paper. Also, authors should include missing references. Plagiarism is not acceptable at all:
  • “In order to analyze the complex spectra from these broadband measurements, an automated spectral assignment software application called AUTOFIT was developed [5]. AUTOFIT was” (page1), “From an algorithmic point of view, the detection of a molecule within an experimental spectrum can be implemented as an exploration of many combinations of triplets. The number of algorithm cycles necessary to the detection purpose varies according to the required accuracy of the goal.” (page4), “that number depends on the size of the frequency window where to seek for transitions, therefore it is not rare that a few thousands or even millions of combinations must be checked out. In its original design, AUTOFIT is capable of analyzing 35-50 triplets per second per CPU core, therefore a typical 4-core processor can approximately process 250 triplets per second. A typical run would take from 2 to 3 hours considering a frequency window of 200-300 MHz. The increase in the window” (page4) … etc.
  • “is considered to scale linearly if the speedup (in terms of work units completed per unit time) is equal to the number of processing elements used (N).” (page7) … etc.

.

.

.

Etc.

  • English writing: This paper requires moderate proofreading (typos and grammar). Authors should accurately check the paper to be free from grammatical and typo mistakes.
  • List of References: It requires updating. Also, the references should follow Electronics-MDPI style. Some references are not necessary and require remove them. Some search names in the reference list begin an uppercase letter for each word and others use only an uppercase letter in the first word, authors should standardize style. Some of the references do not contain enough information such as [31], [33] … etc. Some references are not accessible such as [31]. This paper requires accurately extensive check for list of references to remove all problems.

Author Response

The authors have to accurately address the below comments. 

C4.1: Paper title: It prefers to replace the abbreviations (such as HPS ...) in the search title with words to be clearer to the readers and researchers. 

R4.1: We thank the reviewer for pointing this out. We expanded the HPC acronym in the paper title.   

C4.2: The "Literature review" section contains a lot of useful information, however, the authors point out some references that used "AUTOFIT" but they did not clearly state drawbacks of these papers. 

R4.2: We thank the reviewer for the comment. In Section 2 (Literature review, lines 109-110), we have added a line that touches on AUTOFIT’s main drawback, which is discussed later in Section 3. Specifically, in section 3 we have stressed out the limitation of applicability of AUTOFIT in experiments conducted with technologically advanced spectrometers. 

C4.3: The authors presented their findings in an appropriate structured way, and they also provide some flaws in their proposed nicely. However, the authors did not provide comparisons between their findings and the results of existing research to support rationality and fairness in the superiority of their proposed (despite the different environments and parameters). 

R4.3: We thank the reviewer for the interesting point of discussion. Unfortunately, this question is beyond the scope of this paper and is currently not easily solved. Indeed, in Section 2 “Literature review" we have reported different methods for the analysis of the spectra highlighting the main characteristics of each. However, as reported in the text, the development of these approaches is still at an early stage and there is no comprehensive method that can currently analyze every possible case. Furthermore, for some of these methods (for example the one based on the artificial neural network) the algorithm used is not yet available. In our specific case, AUTOFIT, we have reported the advantages and limitations of the method and this work serves to highlight the progress made with this approach. 

C4.4: Figures and Tables: All figures and tables are drawn in high resolution. However, Figures 4, 5 and Tables 1, 2 shown in-text before use. 

R4.4: Whenever possible, we preferred to display figures/tables on the same page where they are referenced in the text; otherwise, we display them on the very next page (as suggested by the reviewer). If this does not align with the journal guidelines, we are prepared to comply. 

C4.5: Paraphrase text: There are many sentences taken from previous papers. Authors must paraphrase all sentences and phrases taken from the previous papers. The authors must avoid taking whole sentences from the original papers even if it is for the same authors. They are obligated to paraphrase during the revision of the entire paper. Also, authors should include missing references. Plagiarism is not acceptable at all: 

    “In order to analyze the complex spectra from these broadband measurements, an automated spectral assignment software application called AUTOFIT was developed [5]. AUTOFIT was” (page1), “From an algorithmic point of view, the detection of a molecule within an experimental spectrum can be implemented as an exploration of many combinations of triplets. The number of algorithm cycles necessary to the detection purpose varies according to the required accuracy of the goal.” (page4), “that number depends on the size of the frequency window where to seek for transitions, therefore it is not rare that a few thousands or even millions of combinations must be checked out. In its original design, AUTOFIT is capable of analyzing 35-50 triplets per second per CPU core, therefore a typical 4-core processor can approximately process 250 triplets per second. A typical run would take from 2 to 3 hours considering a frequency window of 200-300 MHz. The increase in the window” (page4) … etc. 

    “is considered to scale linearly if the speedup (in terms of work units completed per unit time) is equal to the number of processing elements used (N).” (page7) … etc. 

R4.5: We thank the reviewer for the suggestion. We reviewed the paper and paraphrased sentences taken from our former works. Also, where necessary we added missing references. 

C4.6:    English writing: This paper requires moderate proofreading (typos and grammar). Authors should accurately check the paper to be free from grammatical and typo mistakes. 

R4.6: We thank the reviewer for noticing the presence of errors in the text, so we could further improve the quality of our work. We thoroughly proofread the manuscript to fix typos and errors 

C4.7: List of References: It requires updating. Also, the references should follow Electronics-MDPI style. Some references are not necessary and require remove them. Some search names in the reference list begin an uppercase letter for each word and others use only an uppercase letter in the first word, authors should standardize style. Some of the references do not contain enough information such as [31], [33] … etc. Some references are not accessible such as [31]. This paper requires accurately extensive check for list of references to remove all problems. 

R4.7: We thank the reviewer for pointing this out. We revised the references list in order to comply with the proposed suggestion. The only references containing a title with capitalized letters are AUTOFIT and HS-AUTOFIT, which reflect the actual word provided by authors in their published papers. Also, we re-compiled the latex source file using MDPI suggested templates and bib styles. Finally, we fixed the issue with reference [35] (former [31]). Regarding unnecessary references, we believe all references reported in the bibliography help the reader to catch up with the background and the literature in the field. We kindly ask the reviewer to point out which of them they deem not necessary. 

Round 2

Reviewer 2 Report

Still, I think it's more like a conference-like paper considering the length and technical depth.

However, I agree to accept if the editors are willing to do so (no more review if a revised version is here).

Author Response

C2.1: Still, I think it's more like a conference-like paper considering the length and technical depth. However, I agree to accept if the editors are willing to do so (no more review if a revised version is here). 

R2.1: We are sorry to hear that the reviewer is not satisfied with the paper’s length and technical depth, also considering that other reviewers have not expressed a similar complaint. We point out once again that we are very keen to improve the paper should we receive more specific suggestions.

Reviewer 4 Report

The authors have responded to most of our concerns but there are still some (minor) comments that require addressing.

  • English writing: This paper requires still moderate proofreading (typos and grammar). Authors should accurately check the paper to be free from grammatical and typo mistakes.
  • List of References: It requires still updating. More than half of the research is outdated. Some references are not necessary and should remove them (such as [24] and [27]). Some search names in the reference list begin an uppercase letter for each word (such as [7], [18] … etc.) and others use only an uppercase letter in the first word (such as [5] … etc.), authors should standardize style. This paper requires accurately moderate check for list of references to remove all problems.

Author Response

The authors have responded to most of our concerns but there are still some (minor) comments that require addressing. 

C4.1: English writing: This paper requires still moderate proofreading (typos and grammar). Authors should accurately check the paper to be free from grammatical and typo mistakes. 

R4.1: We proofread the paper once more. We believe it is free of typos/language issues. We kindly ask the reviewer to point out any specific issue they may have spotted. 

C4.2: List of References: It requires still updating. More than half of the research is outdated. Some references are not necessary and should remove them (such as [24] and [27]).  

R4.2: We agree on dropping ref. [24], although it’s very technical and thus supportive of the discussion. Concerning reference [27], it’s the only one reporting on the use of the discussed technique in the field of astrochemistry (actually, the one that mostly deploys it). That reference helps us motivate our work, therefore we'd prefer to keep it. 

C4.3: Some search names in the reference list begin an uppercase letter for each word (such as [7], [18] … etc.) and others use only an uppercase letter in the first word (such as [5] … etc.), authors should standardize style. This paper requires accurately moderate check for list of references to remove all problems. 

R4.3: We thank the reviewer for pointing this out. Now, all references show an uppercase letter only in the first word of the title